# Comparison of Long-Term Skin Quality and Scar Formation in Partial-Thickness Burn Wounds Treated with Suprathel^®^ and epicite^hydro®^ Wound Dressings

**DOI:** 10.3390/medicina58111550

**Published:** 2022-10-28

**Authors:** Jennifer Lynn Schiefer, Friederike Genoveva Aretz, Paul Christian Fuchs, Rolf Lefering, Pouyan Yary, Christian Opländer, Alexandra Schulz, Marc Daniels

**Affiliations:** 1Department of Plastic, Reconstructive, Hand and Burn Surgery, Hospital Cologne Merheim, University of Witten-Herdecke, 58455 Witten, Germany; 2Institute for Research in Operative Medicine (IFOM), Faculty of Health, Witten/Herdecke University, 51109 Cologne, Germany

**Keywords:** Suprathel^®^, epicite^®^, burns, scarring, wound dressing

## Abstract

*Background and Objectives*: Scar formation after burn trauma has a significant impact on the quality of life of burn patients. Hypertrophic scars or keloids can be very distressing to patients due to potential pain, functional limitations, or hyper- or hypopigmentation. In a previous study comparing Suprathel^®^ and the new and cheaper dressing epicite^hydro®^, we were able to show that pain reduction, exudation, and time until wound-healing of partial-thickness burn wounds were similar, without any documented infections. No study exists that objectively measures and compares skin and scar quality after treatment with Suprathel^®^ and epicite^hydro®^ at present. *Materials and Methods*: In this study, the scar quality of 20 patients who had been treated with Suprathel^®^ and epicite^hydro®^ was objectively assessed using the Cutometer^®^, Mexameter^®^, and Tewameter^®^, as well as subjectively with the Patient and Observer Scar Assessment Scale, 3, 6, and 12 months after burn injury. *Results*: In all performed measurements, no significant differences were detected in scar formation after treatment of partial-thickness burn wounds with the two dressings. *Conclusions*: Both the newer and less expensive wound-dressing epicite^hydro®^ and the well-known wound-dressing Suprathel^®^ resulted in stable wound closure and showed good cosmetic results in the follow-up examinations.

## 1. Introduction

Worldwide, burn injuries are the fourth most common type of injury, although incidences are decreasing [1,2,3]. As an example, the incidence in Germany for minor burns is 600/100,000 inhabitants per year, and the incidence for severe burns is 1/60,000 inhabitants per year [4]. Moreover, scarring after burns has a significant impact on patients’ quality of life [5]. Hypertrophic scars and keloids both cause pain and functional impairments, as well as hyper- or hypopigmentation, which can be distressing for patients [5]. Scarring after burn trauma can be influenced by several factors, including burn depth [6], skin color, location of injury, age, and female sex. While these factors cannot be modified by physicians, the choice of appropriate treatment can be modified. Therefore, in recent decades, various wound dressings have been developed for the treatment of superficial burn wounds [7,8,9,10,11,12]. An important aspect when selecting the ideal wound dressing is the reduction in scarring [5,13,14,15,16].

The synthetic caprolactone dressing Suprathel^®^ (PolyMedics Innovations GmbH, Denkendorf, Germany) consists of polylactic acid, trimethylenecarbonate, and e-caprolactone and remains on the wound site until re-epithelialization is complete [17]. Suprathel^®^ has successfully been used in the treatment of partial-thickness burn injuries [15,17,18,19,20]. In a comparison between Suprathel^®^ and Omniderm^®^ (Omikron Scientific Ltd., Rehovot, Israel), a transparent, hydrophilic polyurethane membrane, Schwarze et al. did not observe hypertrophic scarring with both dressings during the 3-month follow-up examination [20]. However, increased patient comfort for the Suprathel^®^ group was described by a significant pain reduction and fewer dressing changes [20,21]. 

In a study conducted by Keck et al., the surface of the Suprathel scar was closer to normal skin compared to areas after skin grafting, which underlines the focus on ensuring a scar-free wound-healing and reducing split-thickness skin grafting [19]. This results in fewer surgeries, less blood loss and the avoidance of painful skin donor sites. In another study, Suprathel^®^ was compared to Mepilex Ag^®^ (Mölnlycke Health Care AG; Göteborg, Sweden), a silver-coated foam dressing, by Hundeshagen et al. [17]. One-month post-burn, they performed an assessment using the Patient and Observer Scar Assessment Scale (POSAS). Patient ratings did not differ for pain, color, itch, stiffness, or irregularity [17]. However, in the Mepilex group, the viscoelasticity of burned skin was increased compared to uninjured skin. In contrast, there was no difference in the Suprathel^®^ group [17]. Rashaan et al. reported good scar quality 6 months post-burn for Suprathel^®^-treated partial-thickness burn wounds [15]. 

Owing to the increasing cost optimization in the healthcare system and the comparably high price of Suprathel^®^ [17,20,22], alternative wound dressings have been developed and the continuous development of wound dressings with similar properties and increased cost-effectiveness is an ongoing process.

Therefore, one of the latest product developments in the field of burn injury is epicite^hydro®^ (QRSKIN GmbH; Würzburg, Germany), which is made of bacterial nanocellulose (BNC) [22]. The nanofiber network was synthesized by *Komagataeibacter xylinus* and consists of 95% water [23,24]. Bacterial nanocellulose is characterized by high biocompatibility [22] and creates a moist environment, supports dermal hydration [22,25], has a cooling effect, reduces intradermal temperature [23] and is loadable with antiseptic solutions [25,26]. Maurer et al. compared bacterial nanocellulose dressings with polyurethane foam dressings in the management of thermal injuries in 92 children and reported scar-free wound reepithelialization for BNC-based wound dressings within 12 days of injury [14]. In addition, they reported fewer dressing changes and documented lesser hypertrophic scarring in the nanocellulose group [14]. In another study, Cattelaens et al. assessed scar quality six months after burns for nanocellulose-based wound dressings in 54 children. They demonstrated normal pigmentation, vascularity, pliability, and normal height of scars [22]. 

In a previous study comparing Suprathel^®^ and epicite^hydro®^, we were able to show that pain reduction, exudation, and time until wound-healing of partial-thickness burn wounds were similar without any documented infections [27]. Nevertheless, no study has objectively compared the skin and scar quality after treatment with Suprathel^®^ and epicite^hydro®^. Therefore, our aim was to assess skin quality and burn scar outcomes at 3, 6, and 12 months after injury in patients who were treated with Suprathel^®^ and epicite^hydro®^ simultaneously.

## 2. Materials and Methods

### 2.1. Patients

Approval from the appropriate ethics committee was obtained 2018 before the inclusion of patients (Ethic approval No.: 5/2018), and the study protocol conformed with the ethical guidelines of the 1975 Declaration of Helsinki. All patients provided written informed consent to participate. Between October 2018 and February 2020, a total of 20 patients aged from 18 to 75 years who sustained partial thickness flames, scalds, or contact burns with more than 0.5% of their total body surface area (TBSA) were enrolled in a prospective, unicentric, open, comparative, intra-individual clinical study, which was performed at the Burn Centre Cologne, Germany. None of these patients had been treated before enrolment with topical agents or pharmaceutical dressings. Furthermore, none of the patients had facial or infected burn wounds. 

### 2.2. Wound Regime

The wound regime was analogous for both dressings. After the wounds were mechanically cleaned using Prontosan^®^ (B. Braun Melsungen AG, Melsungen, Germany), a wound-irrigation solution, and cotton gauze, TBSA and depth assessment were performed by an experienced burn surgeon. Wounds with the same burn depth were divided in two parts and partly covered with Suprathel^®^ and partly covered with epicite^hydro®^. The wound covering was completed with a layer of fatty gauze and an external dressing (Figure 1 and Figure 2). Both dressings remained on the wounds until re-epithelialization was complete, as described in our previous study [27].

### 2.3. Assessment of Scar Quality

The primary outcomes investigated in this study were scar and newly formed skin quality. All patients were requested to participate in follow-up examinations after 3, 6, and 12 months. At these follow-up appointments, scar quality was instrumentally assessed using the following diagnostic devices: Cutometer^®^, Mexameter^®^, Tewameter^®^, (all Courage + Khazaka, Cologne, Germany) and Lightguide Spectrophotometer (O2C^®^, “oxygen to see” by LEA Medizintechnik, Giessen, Germany).

Cutometer^®^ measures the firmness and elasticity of the skin [28,29,30]. The skin’s ability to resist suction (firmness) and to return to its original state after the vacuum is removed (elasticity) is determined. The following values are determined with the help of the cutometer: R0 for skin deformation properties (resistence) and R2 and F1 for skin elasticity; the higher R2 is and the lower F1, the more elastic the skin. 

Mexameter^®^ objectifies the patients’ perception of skin redness and pigmentation rate and is easily reproducible [31,32]. The measurements of melanin (in UNITs) and severity of erythema (in UNITS) in the skin are based on the simple absorption and reflection of light emitted by the skin.

The barrier function of the stratum corneum can be assessed by transepidermal water loss (TEWL), which can be used as an objective evaluation tool for functional abnormalities of the skin, particularly those of the epidermis, and is measured using the Tewameter^®^ [33,34]. The Tewameter^®^ probe indirectly measures the rate of water evaporation (in g/h/m^2^) from the skin through two pairs of sensors (temperature and relative humidity) in a hollow cylinder.

Lightguide spectrophotometry is a diagnostic device for the non-invasive determination of oxygen supply in the microcirculation of blood-perfused tissues [35,36,37]. This measurement method can be used to determine blood flow, oxygen saturation (in %) and hemoglobin levels (in arbitrary units) in captured microvessels. This can determine hypoxia due to venous congestion (low flow, low sO_2_, high rHb), hypoxia due to ischemia (low flow, low sO_2_, low rHb) or hypoxia due to increased metabolism (normal or high flow, low sO_2_, normal rHb).

In addition, scar/newly formed skin quality was assessed by collecting data based on the Vancouver Scar Scale [38,39]. The Vancouver Scar Scale is composed of a point system in which points are distributed in the following categories: Vascularity, Pigmentation, Pliability and Height. A maximum score of 13 can be achieved, which is also the worst scar configuration.

### 2.4. Statistical Methods

Microsoft Excel (Version 16.52, Microsoft, Redmond, WA, USA) was used to manage the data and design the charts. The final analysis was performed using SPSS (Version 26, IBM, Armonk, NY, USA) version 27. An extensive description analysis was performed, with mean (SD) and box plots, with a focus on clinically relevant differences between the two treated areas and the treated and untreated areas. A formal statistical evaluation was not performed due to the limited number of cases and the large number of potential comparisons (11 measurements at 3 locations and 3 timepoints).

## 3. Results

Thirteen male patients and seven female patients (*n* = 20) participated in the scar evaluations after 3,6 and 12 months with a mean age of 36.6 years and a mean TBSA of 9.2% (1–23%). Three patients were older than 50 years old (51, 55 and 61 years), Mean age was 36.6 years (min. 18 years; max. 61 years). The majority of patients were injured by hot liquid (45%), followed by flame burns (30%). 

### 3.1. Skin Elasticity

Scar/newly formed skin quality was analyzed using Cutometer^®^ measurements. As summarized in Table 1, all analyzed parameters showed similar mean and median values for the two treatment groups when compared to each other or to the data of unwounded skin (normal) in the measurements at 3, 6, and 12 months (Figure 3, Table 1).

### 3.2. Skin Pigmentation

As summarized in Table 2, all analyzed parameters showed similar mean and median values for the two treatment groups when compared to each other. Compared to the data of unwounded skin (Normal), treatment with Suprathel^®^ and epicite^hydro®^ showed higher mean values for the skin redness parameter at 3 and 6 months after treatment (Table 2). The values comparing Suprathel^®^ and epicite^hydro®^ did not relevantly differ. Although measurements at 12 months showed further improvements in erythema scores, both treatment groups showed higher scores compared to those of uninjured skin (Figure 4). However, there was no relevant difference between the two wound dressings (Table 2). While melanin levels at the 12-month examination were similar between the two groups compared to those of the uninjured skin, melanin levels were lower in the Suprathel^®^ group than in the uninjured skin, without clinical relevancy.

### 3.3. Trans-Epidermal Water Loss (in g/h/m^2^) 

As summarized in Table 3, all analyzed parameters showed similar mean and median values for the two treatment groups when compared to each other and to the data of unwounded skin (Normal) (Figure 5).

### 3.4. Skin Evaluation with Lightguide Spectrophotometry 

All analyzed parameters showed similar mean and median values for the two treatment groups when compared with each other (Table 4). Compared to the data of unwounded skin (Normal), the data at 3 and 6 months after treatment showed higher values for the sO2 rHb and flow parameters, with the exception of the epicte^hydro^ group for sO2 after 6 months (*p* = 0.707) (Table 4, Figure 6). After 12 months, only the epicite^hydro^ group showed an increased rHb value compared to unwounded skin (Table 4, Figure 6), without clinical relevance. 

Additionally, scar/newly formed skin quality was subjectively assessed by collecting data based on the Vancouver Scar Scale (VSS). 

### 3.5. Vancouver Scar Scale

Scores for pliability and height after 3 and 6 months were generally low. Scores for pigmentation and vascularity were in the midrange, decreasing from 3 to 12 months (Table 5). 

In summary, the majority of collected data showed very low average values that did not show any significant difference or trend of a difference between the treatments with the two dressings.

## 4. Discussion

Scarring plays an important role in the recovery of quality of life and functionality after burn injury [40]. Therefore, the long-term evaluation of scarring and functionality post-burn is very important [41]. In addition to functionality, aesthetic results are also of great importance to patients. Since scar-free healing plays an important role in the selection of the ideal wound-dressing, various wound dressings for the treatment of superficial burn wounds were developed in recent decades [5,7,8,9,13,14,15,16].

To the best of our knowledge, this is the only study comparing the long-term outcomes of burn wound treatment with epicite^hydro®^ and Suprathel^®^. 

In comparison to normal skin, scars show some specific characteristics, such as a different color, texture, relief, or elasticity and a different trans-epidermal water loss [30,34]. In this study, these parameters and wound treatments were effectively evaluated with different assessment tools.

Several research groups have reported that scar properties after superficial burns improved with Suprathel^®^ treatment [17,19,42]. Hundeshagen et al. compared in their trial the treatment of partial-thickness burns with Suprathel^®^ and Mepilex Ag^®^ [17]. Analysis of skin quality and scarring was performed 1-month post burn. Patients in the Suprathel^®^ group rated the overall appearance of the healed wound as being better [17]. Compared with the baseline hydration and TEWL measurements of the burned skin area were elevated for both wound dressings [17]. However, the data from the present study showed similar mean and median TEWL values for the two treatment groups (Suprathel^®^ and epicite^hydro®^), when compared to each other and to the data of unwounded skin 3, 6, and 12 months after treatment. This discrepancy may have occurred due to the early measurements, performed only 1-month after injury by Hundeshagen et al., because increased hydration and TEWL measurements are related to the repair of the dermal barrier, which decreases with time [34]. Furthermore, Hundeshagen et al. reported no significant differences in pigmentation (erythema and melanin) between normal and burned skin [17]. In contrast, in the present study, comparison of the data of unwounded skin and wounded skin after 3, 6, and 12 months after treatment with Suprathel^®^ and epicite^hydro®^ showed slightly higher mean values for the skin redness parameter 3 and 6 months after treatment. Additionally, Hundeshagen et al. described an increased stiffness of burned skin that was treated with MepilexAg^®^ 1-month post burn; this effect was not seen in the present study [17]. In the current study, skin elasticity did not relevantly differ between Suprathel^®^- and epicite^hydro®^-treatment and normal unwounded skin. Moreover, measurements were performed over a longer period of time in our study. A temporary stiffness after 1 month might not be of high importance if the measurements show no further differences after 3 and 6 months. Usually, scarring is complete after 12–24 months. Therefore, the most important measurement was that after 12 months [40]. 

Hakkarainen et al. compared nanocellulose treatment of skin graft donor sites with Suprathel^®^ treatment [43]. In their study, epithelialization under the nanocellulose dressing was reported to be faster compared to that under Suprathel^®^ [43], a phenomenon that was not seen in our former study with simultaneous application of the two dressings [27]. This is a very important issue because studies have shown that the development of hypertrophic burn scars is lower in burn injuries that heal within 21 days [22,44,45]. Cattelaens et al. treated fifty-six burned children with epilepsy and did not report any hypertrophic scars in partial- and full-thickness burn injuries [22]. Six months post-burn, they reported normal pigmentation in 98% of the treated patients, whereas, in the present study, treatment with Suprathel^®^ and epicite^hydro®^ showed significantly higher mean values for the skin redness parameter at 3, 6, and 12 months, which decreased in the measurements from 3 to 12 months. Since Suprathel has been on the market for a much longer time, more studies can be found. Long-term scar evaluation studies are difficult and time consuming. Therefore, we did not find any studies with objective scar evaluation dealing with scar evaluation 12 months after burn-wound treatment with Epicte.

This is the first study to directly compare the two wound-dressings, Suprathel^®^ and epicite^hydro®^, in the treatment of superficial burn wounds. All measurements in the present study did not show any relevant differences between the treatments with the two dressings. During the measurements, there was a steady improvement in scar quality.

In addition, the comparison of the cost effectiveness of Suprathel^®^ with epicite^hydro®^ revealed, based on the material costs, that epicite^hydro^ is more cost-effective [17,20,22]. Depending on individual pricing between the manufacturer and the hospital, Suprathel^®^ was 3.6 times more expensive than epicite^hydro®^ at our hospital. 

This study had several limitations. First, the study group was rather small, comprising only 20 patients. This induced a limited power to identify relevant differences. Multicenter studies can improve the sample size and power of this study. Furthermore, burn depth was assessed only through clinical examination. Laser Doppler Imaging can be used to improve the accuracy of burn depth assessment.

## 5. Conclusions

In all performed measurements, no significant differences in scarring were detected between the two wound dressings after the treatment of partial-thickness burn wounds. The rather new and cheaper Epicite^hydro®^ and Suprathel^®^ both led to a stable wound closure and showed good cosmetic results in the follow-up examinations.

## Figures and Tables

**Figure 1 medicina-58-01550-f001:**
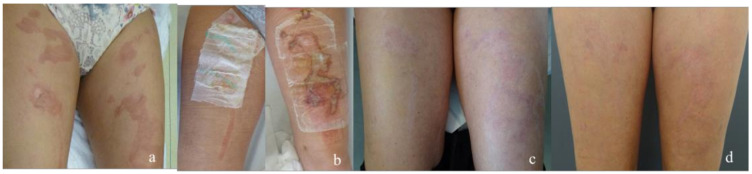
Patient with a superficial burn of the legs. (**a**) burn before debridement of the blisters (**b**) treatment of the left leg with epicite^hydro®^ and the right leg with Suprathel^®^ (**c**) results after 3 months (**d**) results after 6 months.

**Figure 2 medicina-58-01550-f002:**
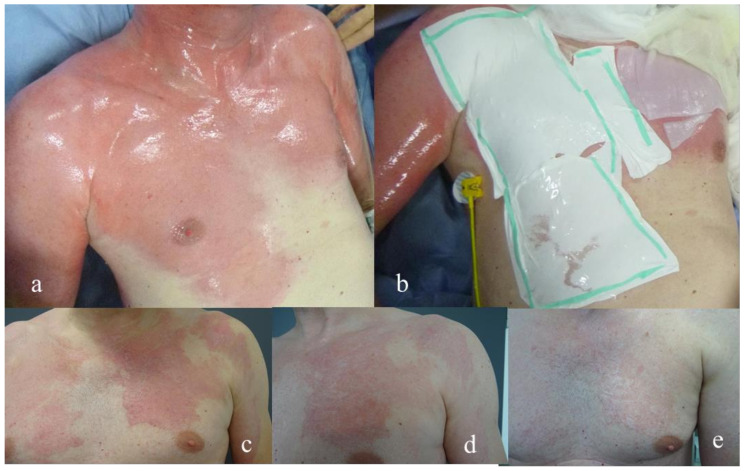
Patient with (**a**) superficial burn of the upper trunk. (**b**) treatment with epicite^hydro®^ and Suprathel^®^ (**c**) results after 3 months (**d**) results after 6 months (**e**) results after 12 months.

**Figure 3 medicina-58-01550-f003:**
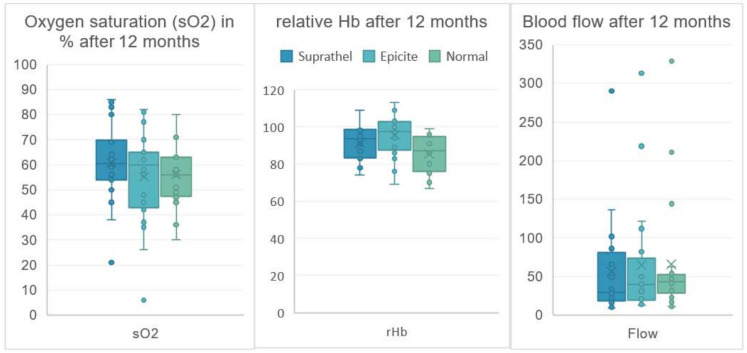
Oxygen Saturation in %, relative Hb and blood flow in Arbitrary Units of treated and untreated areas after 12 months.

**Figure 4 medicina-58-01550-f004:**
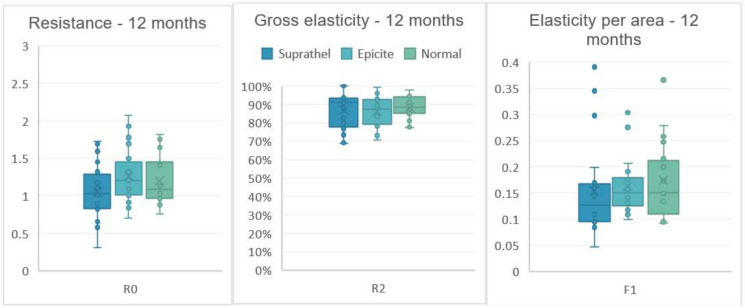
Resistance (R0) in mm, gross elasticity (R2) in % and elasticity per area (F1) in Area under the curve of treated and untreated areas after 12 months.

**Figure 5 medicina-58-01550-f005:**
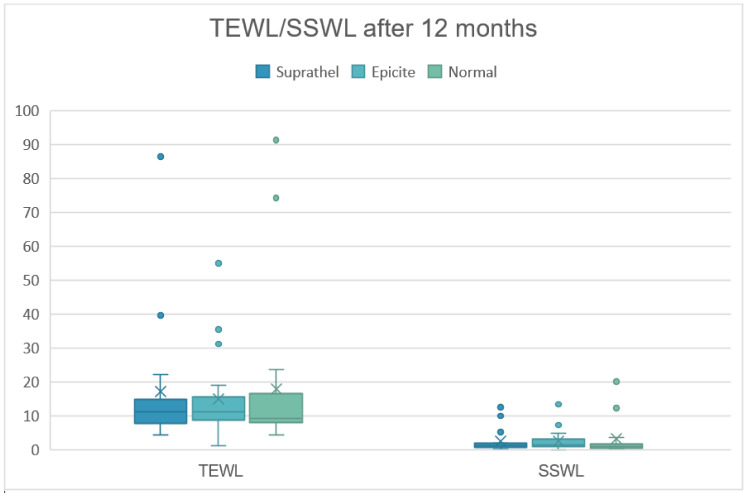
Trans-epidermal water loss (TEWL) and skin surface water loss (SSWL) in g/h/m^2^ of treated and untreated areas after 12 months.

**Figure 6 medicina-58-01550-f006:**
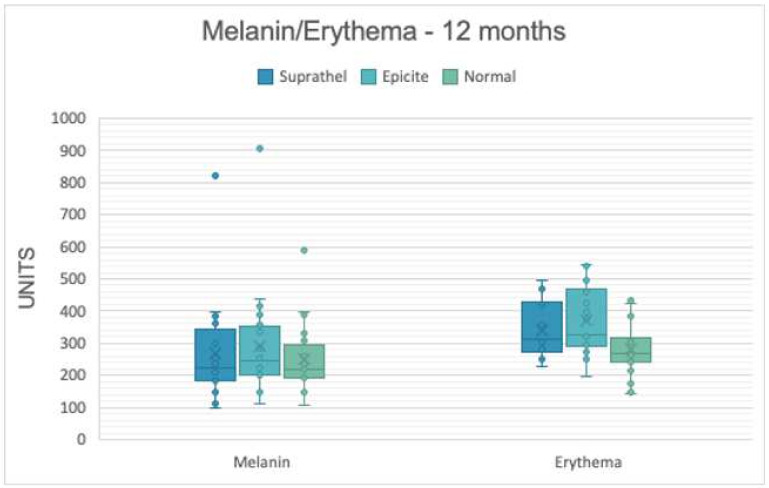
Melanin and erythema of treated and untreated areas after 12 months.

**Table 1 medicina-58-01550-t001:** Cutometer measurements mean and SD (standard derivation) after 3,6 and 12 months of treated and untreated areas R0 in mm; F1 in area under the curve and R2 in %.

Dressing		Suprathel^®^		epicite^hydro®^		Normal	
Cutometer		Mean	SD	Mean	SD	Mean	SD
R0 Resistence	3 months	0.94	0.22	0.99	0.24	1.08	0.25
	6 months	1.06	0.32	1.16	0.36	1.10	0.24
	12 months	1.05	0.37	1.25	0.35	1.20	0.31
R2 Gross Elasticity	3 months	0.83	0.1	0.83	0.11	0.84	0.1
	6 months	0.85	0.11	0.86	0.11	0.87	0.08
	12 months	0.87	0.1	0.85	0.07	0.87	0.05
F1 Elasticity	3 months	0.13	0.06	0.14	0.05	0.14	0.06
	6 months	0.14	0.05	0.17	0.09	0.15	0.05
	12 months	0.14	0.09	0.15	0.04	0.16	0.06

**Table 2 medicina-58-01550-t002:** Mexameter measurements in UNITS in mean and SD (standard derivation) after 3, 6 and 12 months of treated and untreated areas.

Dressing		Suprathel^®^		epicite^hydro®^		Normal	
Mexameter		Mean	SD	Mean	SD	Mean	SD
Melanin	3 months	237.1	125.8	244.6	105.3	251.2	90.7
	6 months	247.7	131.1	268.3	151.4	249.1	102.2
	12 months	265.6	156.5	288.7	170.5	249.3	111.0
Erythema	3 months	378.6	77.1	390.7	89.8	299.0	91.2
	6 months	359.1	85.3	376.3	72.1	270.2	93.6
	12 months	341.3	85.5	368.6	104.6	282.6	86.7

**Table 3 medicina-58-01550-t003:** Tewameter measurements in mean and SD(standard derivation) in g/h/m^2^ after 3,6 and 12 months of treated and untreated areas.

Dressing		Suprathel^®^		epicite^hydro®^		Normal	
Tewameter		Mean	SD	Mean	SD	Mean	SD
TEWL	3 months	18.8	15.5	18.3	16.2	17.8	19.8
	6 months	16.5	11.3	16.2	11.7	13.0	11.6
	12 months	17.1	19.1	15.0	12.4	18.0	22.7
SSWL	3 months	2.0	1.2	2.1	2.2	1.9	2.0
	6 months	2.0	1.6	2.3	2.3	1.6	1.8
	12 months	2.5	3.3	3.1	3.1	3.2	5.4

**Table 4 medicina-58-01550-t004:** O2C measurements in mean and SD (standard derivation) after 3, 6 and 12 months of treated and untreated areas (oxygen saturation sO_2_ in %; relative Hemoglobin (rHb) value and flow in arbitrary units).

Dressing		Suprathel^®^		epicite^hydro®^		Normal	
Oxygen2See		Mean	SD	Mean	SD	Mean	SD
sO_2_	3 months	74.2	11.5	68.1	14.1	58.2	14.1
	6 months	73.3	17.3	61.3	16.0	63.0	12.8
	12 months	61.0	16.2	55.2	19.1	56.1	12.5
rHb	3 months	94.6	11.8	97.5	11.5	82.1	11.2
	6 months	95.2	11.0	97.5	8.5	85.2	10.1
	12 months	91.3	9.1	95.8	11.6	85.3	10.0
Flow	3 months	72.1	57.1	74.3	64.0	57.6	58.0
	6 months	73.2	69.4	77.5	102.9	42.2	30.8
	12 months	57.1	65.7	64.9	76.6	65.6	77.4

**Table 5 medicina-58-01550-t005:** Results of the Vancouver Scar Scale in mean and SD (standard derivation).

Dressing		Suprathel^®^		epicite^hydro®^	
VSS Parameter		Mean	SD	Mean	SD
Pigmen-tation	3 months	1.5	0.607	1.45	0.686
	6 months	1.05	0.887	1.25	0.851
	12 months	0.90	0.912	0.95	0.887
Vasculartity	3 months	1.25	0.444	1.15	0.366
	6 months	0.70	0.657	0.60	0.503
	12 months	0.35	0.587	0.30	0.571
Pliability	3 months	0.00	0.000	0.05	0.224
	6 months	0.15	0.671	0.10	0.447
	12 months	0.30	0.733	0.25	0.639
Height	3 months	0.15	0.366	0.10	0.447
	6 months	0.10	0.308	0.10	0.308
	12 months	0.25	0.550	0.10	0.308

## Data Availability

All data presented in manuscript.

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
