# Peer review of "Comparison of Long-Term Skin Quality and Scar Formation in Partial-Thickness Burn Wounds Treated with Suprathel® and epicitehydro® Wound Dressings"

_medicina, 2022, doi:10.3390/medicina58111550_

Round 1

Reviewer 1 Report

Skin wounds repairment remains a great clinical challenge, and the scarring quality is an important factor affecting the patients’ quality of life. This work presented the long-term outcomes of skin quality and scar formation in partial-thickness burn wounds treated with Suprathel® and epicyte(hydro)® wound dressings. The authors found a stable wound closure, good cosmetic results, and high patient satisfaction in the follow-up examinations. The results indicated a more cost-effective alternative for skin injury treatment. There are some minor concerns to be addressed:

The data in this article appear to be from the identical group of patients, with many same parameters such as n=20, TBSA of 9.2%, 30% of the injury cause was flame burns, etc., in a previously reported clinical trial (likely in Ref. 21), evaluating the same wound dressings. please justify such design that the outcomes were reported separately, in particular the follow-up periods were in the same order of magnitude (12 months versus 6 months). Furthermore, please confirm some differences if the data were obtained from identical patients, e.g., 1-19% vs 1-23% in TBSA.

Please revise inappropriateness in writing or expression, for example, in the last paragraph of the introduction, reference(s) should be provided for the statement: “In a previous study comparing Suprathel® and epicitehydro®, we were able to show that ...” (seemed to be Ref. 21 or anyone else). On line 188 in page 7, “relevantnes” should be “relevance”.

  Please provide excel images with higher resolution, and by the way, the bottom frame in box plots of blood flow after 12 months (Fig.6) seemed to be cut off when taking a screenshot.

Reviewer 2 Report

Dear Author,

Kindly go through each of these points and make the applicable changes to your manuscript

Introduction

1.      It is mentioned that- Scarring after burn trauma can be influenced by several factors, including burn depth [3], dark skin color, location of injury, age, and female sex… kindly consider whether gender can be mentioned instead as different influential factors are enumerated

2.      It is mentioned that- The synthetic dressing Suprathel.. kindly introduce the former further – its composition etc

3.      It is mentioned that- Om niderm® (Omikron Scientific Ltd., Rehovot, Israel),.. kindly mention what does it refer to- a  steroid cream or otherwise so as comparison basis can be comprehensible

4.      It is mentioned that- In a study conducted by Keck et al., the surface of the Suprathel scar was closer to normal skin compared to areas after skin grafting [13]… kindly provide an inference as to the advantages on clinical applications

5.      Dear authors, kindly note that the epicitehydro® description in the introduction ends abruptly. Kindly revise

6.      It is mentioned that- In a previous study comparing Suprathel® and epicitehydro®, we were able to show that pain reduction, exudation, and time until wound healing of partial-thickness burn  wounds were similar without any documented infections… kindly support with reference citation

Methodology

1.      Kindly mention the place of the study in the beginning of the methodology

2.      It is mentioned that- Lack of consent and compliance in the follow-up examinations were among the exclusion criteria, as described in our previous study[21]. Kindly exclude or revise the narration. Kindly revise wherever applicable too

3.      It is mentioned that- Figure 2. Patient with (a) superficial burn of the trunk. Dear authors, kindly comment whether the whole trunk was involved or can it be specific to upper trunk

4.      It is mentioned that- All patients in our previous study were requested to participate in follow-up examinations after 3, 6, and 12 months.

Dear authors, kindly note that the assessment parameters included in the present study are different from the authors previous study as stated. Therefore, the manuscript can have a detailed account pertaining to the study factors considered alone

OR

If the patient’s participation were to be mentioned and related to the authors previous study, kindly consider mentioning it once in beginning of the methodology which suffices

5.      Kindly mention how were the evaluation methods considered were scored. Also, kindly comment how was the site of evaluation decided, how many sites were considered as the figures provided suggest of a scope for considerable area for evaluation

6.      It is mentioned that- and the large number of potential comparisons (11 measurements at 3 locations and 3 time points). Dear authors, kindly comment on the former before the statistical analysis. Kindly enumerate the different measurements considered for measuring the skin quality through Vancouver Scar Scale

7.      Dear authors, kindly note that though the no: of parameters assessed  are large, however methodology needs to be elaborated further  wrt the measurements considered for assessment of scar quality – enumeration ,the mode of scoring, the evaluation or mode of calculation whichever applicable can be mentioned in brief. Kindly consider

Results

1.      It is mentioned that- R0 for skin deformation properties and R2 and F1 for skin elasticity; the higher R2 is and the lower F1, the more elastic the skin).. kindly consider mentioning and discussing further in the methodology instead. Kindly mention how were the said factors were evaluated in a sentence

2.      Kindly verify the R0 designation mentioned both in the table and the graph mentioned

3.      Skin elasticity: kindly consider mentioning at different time intervals too in the text irrespective of the tables and graphs inclusion

4.      It is mentioned that- Trans-epidermal water loss (TEWL) and skin surface water loss (SSWL).. kindly mention how were the former were calculated in methodology very briefly

5.      Dear authors, kindly note as both the treatment groups were analyzed at three different time intervals, kindly revise accordingly in results narration

6.      It is mentioned that- the sO2 – kindly mention what does it signify and also kindly verify the representation

7.      It is mentioned that- HB parameters.. kindly verify the representation and also, the table as rHb is mentioned. Dear authors, kindly verify and be specific

8.      It is mentioned that- relative Hb and relative Haemoglobin value … kindly verify and comment

9.      Kindly structure the narration of results as different time periods were also considered

10.  Kindly mention the inference of the study at the end of the results

Discussion

1.      It is mentioned that- 1-month after injury by Hundeshagen et al. increased hydration and TEWL measurements are related to the repair of the dermal barrier,.. kindly verify the flow

2.      It is mentioned that- this effect was not seen in our study.. kindly refer as the current/ present study instead

Conclusion

1.      It is mentioned that-  leading to high patient satisfaction…. Kindly consider discussing the former in the results and the discussion too

Regards

Reviewer 3 Report

English should be improved, namely in the introduction.

 The authors should avoid copying parts of the manuscript in the abstract.

 References should as recent as possible. Reference number one is from 2011. Could the authors provide a more updated data on the incidence of burr injuries? Another example is in line 39, where the authors make reference of two reviews. Please provide more recent reviews or studies.

Regarding the patients, there are still some unanswered questions:

-How many patients were older than 50?

-The minimum TBSA was 0.5%. What was the maximum?

-Is the covered skin area of Suprathel the same as for Epicite, in all the patients? What body parts were affected?

-How often the dressings were changed? Were both types of dressings changed at the same time point?

-Were the patients hospitalized? What was the duration of the hospitalization?

Discussion:

The authors could provide a hypothesis for the higher level of skin redness presented by epicyte treated skin.

The authors discuss different works of suprathel, but do not seem to discuss/compare the results with other epicyte studies. Please add some discussion/comparison on this.

Minor corrections:

Abstract: add ® to Cutometer, Mexameter, and Tewameter,

Line 36: should be “skin color” instead of “dark skin color”?

Line 37: say “gender” instead of “female sex”

Line 75-77: please insert the reference of the study where the authors have previously compared both dressings.

Figure 1: Could the authors provide a photograph of the patient’s legs before treatment?

Figure 1 does not show the results after 12M

Figure 2 does not show the results after 6M

Tables- could be simplified and better looking by including Mean and SD in the same column, instead of presenting it separately.

Table 5 should follow the same format as the previous ones, such as:

-numbers format

- missing ®

- “after 3 months” should be “3 months”.
